# Glucose-6-phosphate dehydrogenase activity in individuals with and without malaria: Analysis of clinical trial, cross-sectional and case–control data from Bangladesh

Benedikt Ley[1]*, Mohammad Shafiul Alam[2], Mohammad Golam Kibria[2], Jutta Marfurt[1], Ching Swe Phru[2], Jenifar Quaiyum Ami[2], Kamala Thriemer[1], Sarah Auburn[1], Nusrat Jahan[2], Fatema Tuj Johora[2], Mohammad Sharif Hossain[2], Cristian Koepfli[3], Wasif Ali Khan[2], Ric N. Price[1,4,5]

1 Global and Tropical Health Division, Menzies School of Health Research and Charles Darwin University, Darwin, Northern Territory, Australia, 2 Infectious Diseases Division, International Centre for Diarrhoeal Disease Research, Bangladesh, Dhaka, Bangladesh, 3 Eck Institute for Global Health, Department of Biological Sciences, University of Notre Dame, Notre Dame, IN, United States of America, 4 Mahidol Oxford Tropical Medicine Research Unit, Faculty of Tropical Medicine, Mahidol University, Bangkok, Thailand, 5 Centre for Tropical Medicine and Global Health, Nuffield Department of Clinical Medicine, University of Oxford, Oxford, United Kingdom

* Benedikt.Ley@menzies.edu.au

**Data Availability Statement:** All relevant data are included as supporting information.

## Abstract

### Background

Glucose-6-phosphate dehydrogenase (G6PD) activity is dependent upon *G6PD* genotype and age of the red blood cell (RBC) population, with younger RBCs having higher activity. Peripheral parasitemia with *Plasmodium* spp. induces hemolysis, replacing older RBCs with younger cells with higher G6PD activity. This study aimed to assess whether G6PD activity varies between individuals with and without malaria or a history of malaria.

### Methods and findings

Individuals living in the Chittagong Hill Tracts of Bangladesh were enrolled into 3 complementary studies: (i) a prospective, single-arm clinical efficacy trial of patients (*n* = 175) with uncomplicated malaria done between 2014 and 2015, (ii) a cross-sectional survey done between 2015 and 2016 (*n* = 999), and (iii) a matched case–control study of aparasitemic individuals with and without a history of malaria done in 2020 (*n* = 506). G6PD activity was compared between individuals with and without malaria diagnosed by microscopy, rapid diagnostic test (RDT), or polymerase chain reaction (PCR), and in aparasitemic participants with and without a history of malaria. In the cross-sectional survey and clinical trial, 15.5% (182/1,174) of participants had peripheral parasitemia detected by microscopy or RDT, 3.1% (36/1,174) were positive by PCR only, and 81.4% (956/1,174) were aparasitemic. Aparasitemic individuals had significantly lower G6PD activity (median 6.9 U/g Hb, IQR 5.2–8.6) than those with peripheral parasitemia detected by microscopy or RDT (7.9 U/g Hb, IQR 6.6–9.8, *p* < 0.001), but G6PD activity similar to those with parasitemia detected by

**Funding:** This study was funded by the Bill and Melinda Gates Foundation (OPP1054404) and the Wellcome Trust (Senior Fellowship in Clinical Science awarded to RNP, 200909). SA is supported by a Georgina Sweet Award for Women in Quantitative Biomedical Science and KT is CSL Centenary fellow. No funding bodies had any role in study design, data collection and analysis, decision to publish, or preparation of the manuscript.

**Competing interests:** The authors have declared that no competing interests exist.

**Abbreviations:** AMM, adjusted male median; DSS, demographic surveillance system; G6PD, glucose-6-phosphate dehydrogenase; G6PDd, glucose-6-phosphate dehydrogenase deficiency; Hb, hemoglobin; icddr,b, International Centre for Diarrhoeal Disease Research, Bangladesh; OR, odds ratio; PQ, primaquine; RBC, red blood cell; RDT, rapid diagnostic test; TQ, tafenoquine.

PCR alone (submicroscopic parasitemia) (6.1 U/g Hb, IQR 4.8–8.6, $p = 0.312$). In total, 7.7% (14/182) of patients with malaria had G6PD activity < 70% compared to 25.0% (248/992) of participants with submicroscopic or no parasitemia (odds ratio [OR] 0.25, 95% CI 0.14–0.44, $p < 0.001$). In the case–control study, the median G6PD activity was 10.3 U/g Hb (IQR 8.8–12.2) in 253 patients with a history of malaria and 10.2 U/g Hb (IQR 8.7–11.8) in 253 individuals without a history of malaria ($p = 0.323$). The proportion of individuals with G6PD activity < 70% was 11.5% (29/253) in the cases and 15.4% (39/253) in the controls (OR 0.7, 95% CI 0.41–1.23, $p = 0.192$). Limitations of the study included the non-contemporaneous nature of the clinical trial and cross-sectional survey.

## Conclusions

Patients with acute malaria had significantly higher G6PD activity than individuals without malaria, and this could not be accounted for by a protective effect of G6PD deficiency. G6PD-deficient patients with malaria may have higher than expected G6PD enzyme activity and an attenuated risk of primaquine-induced hemolysis compared to the risk when not infected.

## Author summary

### Why was this study done?

- Primaquine and tafenoquine are the only drugs that prevent relapses of the vivax malaria parasite.

- Primaquine and tafenoquine cause significant side effects in patients with low activity of the G6PD enzyme.

- Approximately 400 million people worldwide are affected by low G6PD enzyme activity.

### What did the researchers do and find?

- We measured G6PD activity among individuals with and without malaria.

- We found G6PD activity to be significantly higher among malaria patients.

- We then enrolled healthy individuals with and without a history of malaria to assess whether people with low G6PD activity were less likely to contract malaria.

- G6PD activity did not differ significantly between people with and without a history of malaria who were free of malaria at the time point of G6PD measurement.

### What do these findings mean?

- We conclude that G6PD activity increases during a malaria infection and that the observed clinically relevant difference in G6PD activity between those with and without

malaria is not due to people with lower G6PD activity being less likely to contract malaria.

- These findings suggest that standard treatment for vivax malaria may be safer than assumed.

- These findings will need to be confirmed.

## Introduction

Over the last 2 decades, significant progress has been made in reducing the burden of malaria. However, the impact of intense malaria control activities has been far greater for *Plasmodium falciparum* than *P. vivax*, which is now the predominant malaria species in many co-endemic countries [1,2]. Unlike *P. falciparum*, *P. vivax* forms dormant liver stages (hypnozoites) that can reactivate weeks to months after the first infection, causing recurrent episodes of febrile illness associated with significant morbidity and mortality [3]. To date, the only drugs that can kill hypnozoites are the 8-aminoquinolines compounds, primaquine (PQ), and tafenoquine (TQ), which, although generally well tolerated, can cause severe hemolysis in individuals with glucose-6-phosphate dehydrogenase (G6PD) deficiency [4]. In view of the risk of severe drug-induced hemolysis, the World Health Organization (WHO) recommends that all patients should be tested for G6PD deficiency (G6PDd) prior to prescribing PQ or TQ [5].

G6PDd is an inherited X-linked enzymopathy with highest prevalence in malaria endemic settings [6,7]. The strong geographic overlap between current and historic areas of malaria transmission and G6PDd suggest an interaction between these exposures; however, investigations of this interaction are conflicting [8]. The correlation between G6PD genotype and phenotype is poor [9–12], since G6PD activity is also determined by a range of factors including the age of the red blood cell (RBC) population, with younger RBCs exhibiting higher levels of G6PD activity than older ones, and, in heterozygous females, the degree of Lyonization [13].

In the presence of oxidative stress, triggered by certain foods, drugs, and infection, G6PD-deficient RBCs hemolyze to a far greater degree than G6PD-normal RBCs [14]. Plasmodia are intraerythrocytic parasites that induce intravascular hemolysis, and this occurs in both parasitized and unparasitized RBCs [15,16]. Whilst *P. vivax* preferentially invades younger RBCs, hemolysis of non-parasitized cells may be particularly marked in older RBCs with lower G6PD activity [17]. We hypothesize that G6PD activity is elevated in patients with malaria compared to individuals without malaria, that this enzyme elevation is attributable to acute parasitemia, and that this enzyme elevation occurs to a greater degree than that attributable to inherent protection from infection in these individuals. The aim of this study was to determine whether G6PD activity varied in individuals with and without malaria and whether this was attributable to a reduced risk of malaria.

## Methods

Malaria in Bangladesh is highly seasonal, with more than 80% of all cases reported during the rainy season from May to October, and 80% to 90% of all reported cases are attributed to *P. falciparum*, with the majority of other cases due to *P. vivax* [18].

Individuals were enrolled into 3 studies at the same location in the Chittagong Hill Tracts, Bangladesh: a clinical trial of patients presenting with uncomplicated *P. falciparum* or *P. vivax*

malaria [19], a cross-sectional survey [20], and a case–control study. First, we determined whether G6PD activity differed between patients with symptomatic malaria, recruited from the efficacy trial, and afebrile and malaria-negative participants from the cross-sectional survey. We then compared G6PD activities between participants with and without a history of malaria, all of whom were afebrile and free of malaria at the time point of G6PD measurement. This study is reported as per the Strengthening the Reporting of Observational Studies in Epidemiology (STROBE) guideline (S1 File).

## Clinical trial

The prospective, single-arm clinical trial [19] was conducted between September 2014 and February 2015 in Alikadam in the district of Bandarban (Chittagong Hill Tracts) and was registered with ClinicalTrials.org as NCT02389374. Patients with uncomplicated malaria diagnosed by microscopy were enrolled and treated according to national treatment guidelines. Venous blood samples were collected for phenotypic and genotypic G6PD testing on day of enrollment, prior to administration of PQ, which was commenced 48 hours later.

## Cross-sectional survey

The cross-sectional survey [20,21] was conducted in the area around Alikadam, district of Bandarban, between August 2015 and January 2016 to assess the prevalence of G6PDd in the local population. Venous blood, collected from 1 household member of randomly selected households, was analyzed for peripheral parasitemia and quantified by microscopy and malaria rapid diagnostic test (RDT) (FalciVax, Zephyr Biomedicals, Goa, India). Parasite species was confirmed by PCR [20]. G6PD status was determined phenotypically and genotypically as described below. Participants with microscopic parasitemia provided a sample for laboratory analysis prior to being referred to the closest healthcare facility, where they were treated according to national treatment guidelines.

## Case–control study

In 2009, a prospective demographic surveillance system (DSS) was established in the Bandarban district amongst a population of 20,558 and is currently part of the INDEPTH network [22–24]. Coordinates of every household within the surveillance area were recorded, and all households were visited between 2009 and 2010 to enroll participants [25]. All participants were assigned a unique identifier (ID) at the initial visit and then reviewed every 3 months, until December 2014, then every 4 months in 2015, and biannually since 2016. At the end of 2018, a total of 19,117 people were under surveillance. At each follow-up visit, a history of malaria was recorded, along with in- and outmigration, births, deaths, and changes of address. Febrile participants were asked to notify the study clinical staff, who then provided a malaria RDT (FalciVax, Zephyr Biomedicals, Goa, India). RDT-positive participants were prescribed free antimalarial treatment according to national treatment guidelines, the episode of malaria was recorded, and the participant was reviewed at the end of treatment to ensure clinical recovery.

This cohort was used to generate a case–control study comprising individuals who were all afebrile and aparasitemic at the time of enrollment. Cases were selected randomly from DSS participants who had had at least 1 episode of malaria confirmed by microscopy during their monitoring within the DSS. Controls were selected from participants with no recorded episodes of malaria during their monitoring within the DSS. Cases and controls were selected in a ratio of 1:1, and matched according to the duration under surveillance, sex, age, and village of residence.

The recorded history of malaria was confirmed by all participants before enrollment, and the absence of parasitemia was determined by RDT (CareStart Malaria HRP2/pLDH [Pf/Pv] Combo RDT, Access Bio, Somerset, NJ, US) and confirmed by microscopic blood film examination. Thick and thin microscopy films were stained with Giemsa, parasites were counted per 200 white blood cells or 2,000 RBCs by 2 independent readers, and the mean of both readings was recorded. In the case of discrepant results, a third reading was done by an expert microscopist, and this finding was considered final.

After informed written consent, individuals were included in the study if no peripheral parasitemia was detected by either RDT or microscopy, and they had an axillary temperature of less than 37.5˚C and no history of fever in the preceding 48 hours. Individuals not fulfilling these criteria were excluded and replaced by participants fitting all criteria until the desired sample size was reached. A venous blood sample was collected from all participants to measure hemoglobin (Hb) concentration, measure G6PD activity by spectrophotometry, and genotyping of the *G6PD* variant.

## Hb measurement and G6PD phenotypic testing

Hb concentration was measured immediately after venipuncture, using a HemoCue 301 (Angelholm, Sweden). Venous samples were stored at 4˚C, and a cold chain was maintained until samples reached the reference laboratory at the International Centre for Diarrhoeal Disease Research, Bangladesh (icddr,b) in the capital Dhaka.

G6PD activity was measured using a UV-1800 Shimadzu spectrophotometer (Shimadzu, Kyoto, Japan). Samples from the efficacy trial and the cross-sectional survey were tested using kits and controls (normal and deficient) from Randox (Crumlin, UK) at 37˚C [19,26]. Samples from the case–control study were analyzed at 30˚C using kits from Pointe Scientific (Canton, MI, US); quality control was done using normal, intermediate, and deficient controls from Analytical Control Systems (Fishers, IN, US) [19,21]. Controls were run daily before testing, which only commenced if the control readings were within the recommended G6PD activity range. All measurements were run in duplicate and repeated if the difference between the first and second measurement was greater than 10%. Since absolute values between different spectrophotometry assays can vary, results from the Randox and Pointe Scientific assays were not compared directly but rather used to categorize participants according to the percentage of normal enzyme activity [27].

## *G6PD* genotypic testing

All samples from patients with G6PD activity less than 70% of the adjusted male median (AMM) [28] and a randomly selected subset of samples with higher G6PD activity were genotyped by PCR at a reference laboratory in Dhaka for the variants Mahidol, Viangchan, Mediterranean, and Orissa, as described previously [29]. All PCR products were also sequenced to identify the Kalyan–Kerala variant [20,30].

## Sample size

G6PD data were available from 175 patients with uncomplicated malaria enrolled into the efficacy trial [19] and 999 individuals from the cross-sectional survey [20,26]. The case–control study was powered to determine a minimal difference of 0.75 U/g Hb between cases and controls with a 95% 2-sided confidence interval and 80% power, which required 253 cases and 253 controls. The total sample size of the 3 studies was 1,680 participants.

### Statistical analysis

Clinical and laboratory data were entered on paper-based case report forms and digitalized using EpiData version 3.1 (EpiData, Odense, Denmark) [31]. All analyses were done using Stata version 14 (StataCorp, College Station, TX, US).

G6PD activity values, measured by spectrophotometry (in U/dl), were divided by the corresponding Hb measurement (in g/dl) collected in the field to provide a result in U/g Hb. The results from Randox control samples generated from the efficacy trial and cross-sectional survey were compared to assess whether the measurements from the trial and survey were comparable.

The AMM was calculated for each spectrophotometry assay as described previously and defined as 100% activity [28]. Current treatment guidelines recommend that individuals with G6PD activity < 30% be categorized as G6PD deficient, those with G6PD activity from 30% to <70% as G6PD intermediate, and those with activity ≥70% as G6PD normal [5,32,33].

Peripheral parasitemia was defined according to 3 categories: (i) individuals with peripheral parasitemia diagnosed by blood film examination or RDT and confirmed by PCR (microscopic parasitemia), (ii) individuals with submicroscopic parasitemia diagnosed by PCR but not microscopy or RDT (submicroscopic parasitemia), and (iii) individuals without detectable peripheral parasitemia assessed by microscopy, RDT, or PCR (aparasitemia).

Normally distributed continuous data were compared using Student *t* test or paired Student *t* test. Non-normally distributed continuous data were compared using the Wilcoxon signed-rank test, the Hodges–Lehmann estimator, or the Kruskal–Wallis equality-of-populations rank test. Unpaired categorical data were compared using the chi squared test or Fisher exact test as appropriate. For each spectrophotometry assay, a multivariable regression model was developed to predict non-normalized G6PD activity (in U/dl), including Hb, sex, and *G6PD* genotype as biologically plausible independent variables. In addition, malaria status (defined as negative, positive only by PCR, or positive by microscopy or RDT) was included in the model comparing participants with and without parasitemia, while history of malaria was included as a predictor in the model corresponding to the case–control study.

### Ethics

All protocols were approved by the Ethics Review Committee of the icddr,b, Bangladesh, and the Human Research Ethics Committee of the Northern Territory Department of Health and Menzies School of Health Research, Australia. Protocol-specific identification numbers together with the exact title of the protocol can be found in S1 Table. Written informed consent was collected from all participants before enrollment, and, in addition, written assent was collected from all minors above the age of 11 years.

## Results

### Cross-sectional survey and clinical trial

Quality control samples were run on 75 normal and deficient control isolates during the efficacy trial and 255 control isolates during the cross-sectional survey. The median reading from normal controls in the efficacy trial was 8.7 U/g Hb (IQR 8.2–9.2) compared to 8.3 U/g Hb (IQR 7.8–8.7) in the cross-sectional survey (difference 0.4 U/g Hb, 95% CI 0.2–0.6). In deficient controls the median was 1.3 U/g Hb (IQR 1.0–1.7) in the efficacy trial and 1.6 U/g Hb (IQR 1.4–1.8) in the cross-sectional survey (difference −0.3 U/g Hb, 95% CI −0.4 to −0.2). After stratifying all results for type of control, a Hodges–Lehmann estimation showed no significant difference between the controls of the 2 studies ($p = 0.962$) (Fig 1).

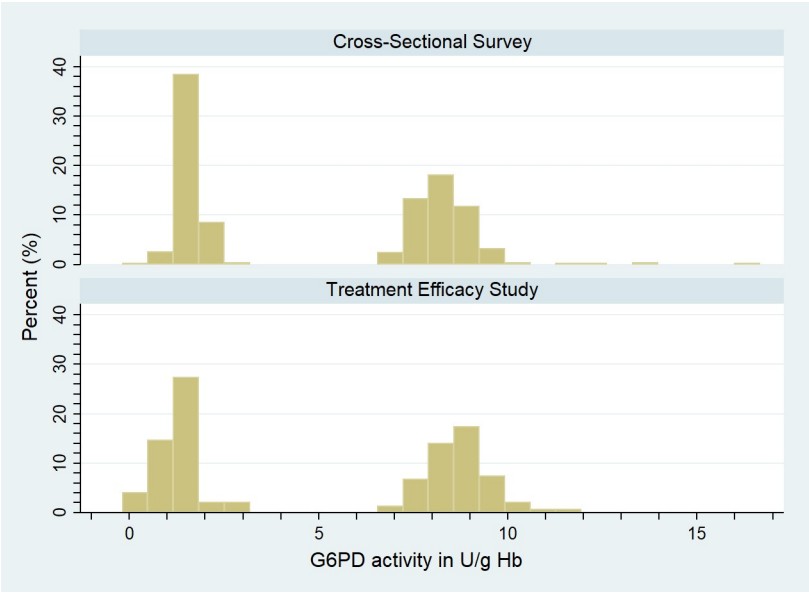

**Fig 1. Distribution of G6PD activity in normal and deficient controls by study.** G6PD, glucose-6-phosphate dehydrogenase.

A total of 1,174 participants were enrolled in the clinical trial and cross-sectional survey and assessed by spectrophotometry, of whom 182 (15.5%) had microscopic parasitemia, 36 (3.0%) had submicroscopic parasitemia, and 956 (81.4%) were aparasitemic (Fig 2). Of the 182

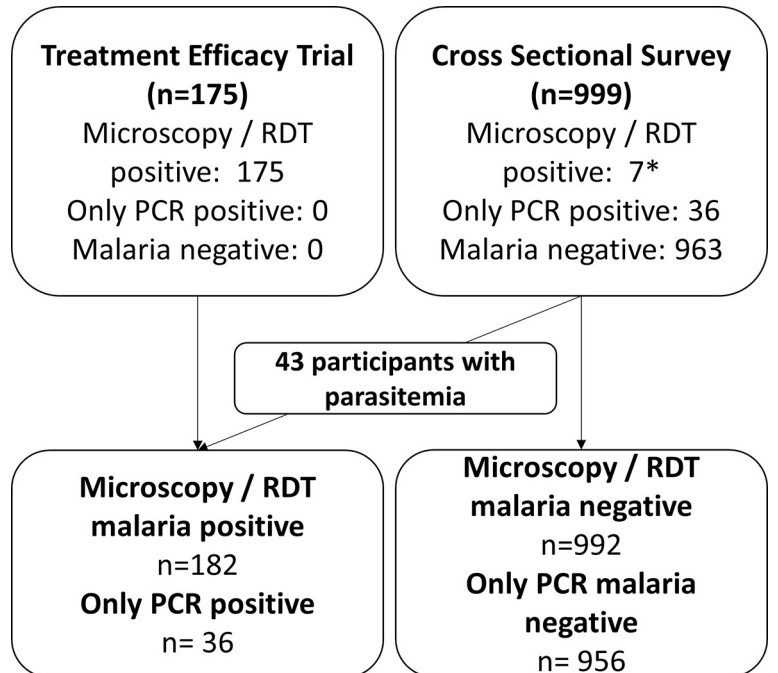

**Fig 2. Study profile of participants enrolled in the treatment efficacy trial and cross-sectional survey.** *Four participants had a *P. falciparum* parasitemia detected by RDT, but not by microscopy. PCR, polymerase chain reaction; RDT, rapid diagnostic test.

**Table 1. Baseline variables stratified by study, and by microscopic malaria and submicroscopic malaria, for participants of the treatment efficacy trail and cross-sectional survey.**

**Comparing studies**

| Variable | Clinical trial (CT, $n$ = 175) | Cross-sectional survey (CSS, $n$ = 999) | Difference (CT minus CSS) | $p$-Value |
|---|---|---|---|---|
| Females, $n$ (%) | 45 (25.7%) | 598 (59.9%) | −34.2% | **<0.001** |
| Mean age, years (95% CI) | 22.0 (20.1–23.9) | 32.5 (31.4–33.6) | −10.5 (−13.3 to −7.8) | **<0.001** |
| Mean body weight, kg (95% CI) | 41.4 (39.1–43.7) | 44.3 (43.6–45.0) | −2.9 (−4.9 to −0.8) | **0.002** |
| Mean body temperature, ˚C (95% CI) | 37.8 (37.6–37.9) | 36.2 (36.2–36.3) | −1.5 (−1.6 to −1.4) | **<0.001** |
| Mean Hb concentration, g/dl (95% CI) | 12.6 (12.3–12.9) | 13.0 (12.9–13.1) | −0.4 (−0.7 to −0.1) | **0.002** |
| Severe anemia (Hb < 7 g/dl), $n$ | 0 | 6 | −6 | 0.379 |
| Geometric mean parasitemia, $\mu l^{-1}$ (95% CI; $n$) | •P. falciparum: 5,692 (3,745–8,651; 111) • P. vivax: 2,661 (1,675–4,228; 54) • P. falciparum + P. vivax mixed: 9,978 (1,279–77,844; 10) | •P. falciparum: 3,986 (77–206,902; 3) | NA | — |

**Comparing participants with parasitemia confirmed by microscopy versus participants with no parasitemia**

| Variable | Microscopic parasitemia (MP, $n$ = 182) | No parasitemia (NP, $n$ = 956) | Difference (MP minus NP) | $p$-Value |
|---|---|---|---|---|
| Females (%), $n$ (%) | 48 (26.4%) | 579 (60.6%) | −34.2% | **<0.001** |
| Mean age, years (95% CI) | 21.6 (19.8–23.5) | 32.8 (31.7–33.9) | −11.2 (−13.9 to −8.5) | **<0.001** |
| Mean body weight, kg (95% CI) | 41.0 (38.7–43.3) | 44.5 (43.8–45.3) | −3.5 (−5.5 to −1.5) | **<0.001** |
| Mean body temperature, ˚C (95% CI) | 37.7 (37.6–37.9) | 36.2 (36.2–36.3) | −1.5 (−1.6 to −1.4) | **<0.001** |
| Mean Hb concentration, g/dl (95% CI) | 12.6 (12.3–12.9) | 13.0 (12.9–13.1) | −0.5 (−0.7 to −0.2) | **<0.001** |
| Severe anemia (Hb < 7 g/dl), $n$ (%) | 0 | 6 (0.6%) | −6 | 0.597 |
| Geometric mean parasitemia, $\mu l^{-1}$ (95% CI; $n$) | •P. falciparum: 5.639 (3,744–8,492; 114) • P. vivax: 2,661 (1,675–4,228; 54) • P. falciparum + P. vivax mixed: 9,978 (1,279–77,843; 10) | — | NA | — |

**Comparing participants with parasitemia confirmed by PCR versus participants with no parasitemia**

| Variable | PCR positive only (PCRP, $n$ = 36) | No parasitemia (NP, $n$ = 956) | Difference (PCRP minus NP) | $p$-Value |
|---|---|---|---|---|
| Females (%), $n$ (%) | 16 (44.4%) | 579 (60.6%) | −16.2% | 0.053 |
| Mean age, years (95% CI) | 29.6 (23.7–35.5) | 32.8 (31.7–33.9) | −3.2 (−9.1 to 2.7) | 0.145 |
| Mean body weight, kg (95% CI) | 41.7 (37.3–46.1)) | 44.5 (43.8–45.3) | −2.8 (−6.7 to 1.1) | 0.080 |
| Mean body temperature, ˚C (95% CI) | 36.4 (36.3–36.6) | 36.2 (36.2–36.3) | 0.2 (0.0 to 0.3) | **0.012** |
| Mean Hb concentration, g/dl (95% CI) | 13.1 (12.7–13.5) | 13.0 (12.9–13.1) | 0.1 (−0.4 to 0.7) | 0.354 |
| Severe anemia (Hb < 7 g/dl), $n$ | 0 | 6 | −6 | 0.801 |

Significant $p$-values given in bold.

NA = not applicable.

participants with microscopic parasitemia, 117 (64.3%) were infected with *P. falciparum*, 54 (29.7%) with *P. vivax*, and 11 (6.0%) with a mixed *P. falciparum* and *P. vivax* infection. Among the 36 participants with submicroscopic parasitemia, 17 (47.2%) had a *P. falciparum* infection, 16 (44.4%) had a *P. vivax* infection, and 3 (8.3%) were positive for both species.

Overall, 25.2% (134/531) of males had microscopic parasitemia, compared to 7.5% (48/643) of females ($p < 0.001$). All subsequent analyses were therefore stratified by sex. Participants in the treatment efficacy study differed significantly from those in the survey: They were younger, had lower body weight, were more likely to be febrile, and had lower Hb concentration (Table 1).

Individuals with microscopic parasitemia had a median G6PD activity of 7.86 U/g Hb (IQR 6.62–9.84), compared to 6.88 U/g Hb (IQR 5.16–8.59) in aparasitemic participants ($p < 0.001$).

**Table 2. Distribution of median G6PD activity stratified by sex and malaria status.**

| Malaria status | Median G6PD activity in U/g Hb (IQR), *n*, *p*-value | | |
| --- | --- | --- | --- |
| | **Males** | **Females** | **All** |
| Aparasitemic (reference) | 6.67 (4.94–8.54), *n* = 377 | 7.01 (5.36–8.67), *n* = 579 | 6.88 (5.16–8.59), *n* = 956 |
| Subpatent parasitemia | 6.84 (4.35–9.45), *n* = 20, *p* = 0.975* | 6.08 (4.92–6.87), *n* = 16, *p* = 0.146* | 6.09 (4.79–8.62), *n* = 36, *p* = 0.312* |
| Microscopic parasitemia | 7.82 (6.59–10.00), *n* = 134, *p* < 0.001* | 8.06 (6.89–9.53), *n* = 48, *p* < 0.001* | 7.86 (6.62–9.84), *n* = 182, *p* < 0.001* |

*Level of significance compared to aparasitemic reference.

G6PD = glucose-6-phosphate dehydrogenase; *n* = number of participants.

Conversely, there was no significant difference in G6PD activity between submicroscopic and aparasitemic individuals (*p* > 0.05) (Table 2; Figs 3 and S1).

The AMM was defined as 7.4 U/g Hb (IQR 5.68–9.07), based on 496 normal males. A total of 77.7% (912/1,174) of participants were categorized as G6PD normal (≥70%), 14.3% (168/1,174) had intermediate G6PD activity (30% to <70%), and 8.0% (94/1,174) were G6PD deficient (<30%).

There was no significant difference in G6PD status between aparasitemic and subpatent parasitemic individuals (Table 2), and these groups were therefore pooled (Table 3). Overall, 9.2% (92/992) of those without microscopic parasitemia had G6PD activities below 30%, compared to 1.1% (2/182) of individuals with microscopic parasitemia (odds ratio [OR] 9.2, 95% CI 2.43–77.71, *p* < 0.001). At the 70% G6PD threshold, 33.3% (244/992) of participants without microscopic parasitemia were G6PD deficient compared to 7.6% (14/182) of patients with microscopic parasitemia (OR = 4.0, 95%CI: 2.26 to 7.61, *p*<0.001) (Table 2). This difference was particularly apparent in females, of whom 23.0% (137/595) with submicroscopic

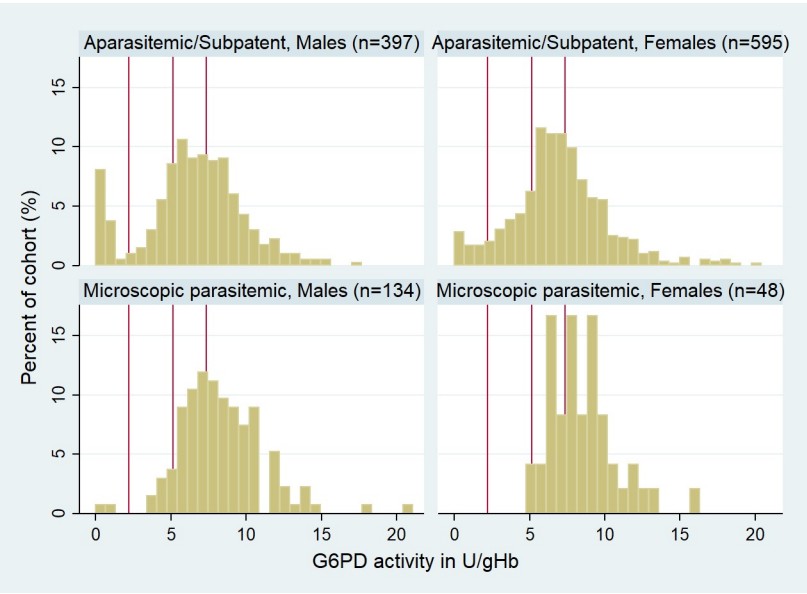

**Fig 3. G6PD activity (in U/g Hb) distribution stratified by microscopic parasitemia and sex in the efficacy trial and cross-sectional survey.** Red vertical lines indicate 30%, 70%, and 100% G6PD activity of the adjusted male median (from left to right). 100% G6PD activity = 7.4 U/g Hb. Participants with submicroscopic parasitemia included in malaria-negative cohort. G6PD, glucose-6-phosphate dehydrogenase.

**Table 3. Proportions of phenotypic G6PD deficient, intermediate, and normal participants among individuals with no parasitemia, parasitemia confirmed by PCR only, and parasitemia confirmed by microscopy.**

| G6PD category | Aparasitemia, n (%) | Subpatent parasitemia, n (%) | Microscopic parasitemia, n (%) | Total n (%) |
|---|---|---|---|---|
| G6PD deficient | 88 (9.2) | 4 (11.1) | 2 (1.1) | 94 (8.0) |
| G6PD intermediate | 149 (15.6) | 7 (19.4) | 12 (6.6) | 168 (14.3) |
| G6PD normal | 719 (75.2) | 25 (69.4) | 168 (92.3) | 912 (77.7) |
| Total | 956 (100.0) | 36 (100.0) | 182 (100.0) | 1,174 (100.0) |

G6PD, glucose-6-phosphate dehydrogenase.

parasitemia or aparasitemia were intermediate or severely deficient compared to 2.1% (1/48) of those with microscopic parasitemia (OR 14.1, 95% CI 2.4–570.9, $p < 0.001$). The corresponding proportions for males were 9.7% (13/134) and 28.0% (111/397) (OR 3.6, 95% CI 1.9–7.3, $p < 0.001$).

In the 178 individuals with microscopic parasitemia, the geometric mean parasitemia was 5,107 (95% CI 3,632–7,180) parasites $\mu l^{-1}$. There was no significant correlation between peripheral parasitemia and G6PD activity ($r_s = 0.001$, $p = 0.988$), and this did not change after stratifying by species (all $p > 0.05$).

Genotyping was undertaken in 29.6% (348/1,174) of patients, including all 94 patients with G6PD activity < 30%, 79.8% (134/168) of randomly selected patients with intermediate G6PD activity, and 13.2% (120/912) of randomly selected G6PD-normal patients. A known *G6PD* variant was identified in 31.6% (110/348) of samples; variants showed a considerable range of G6PD activity (Tables 4 and S2).

The median non-normalized G6PD activity was 90.8 U/dl (IQR 72.4–111.5). In a multivariable regression model (F[9, 338] = 24.78, $p < 0.001$, adjusted $R^2 = 0.3815$) including all 348 participants who were genotyped, the following independent factors were correlated negatively

**Table 4. *G6PD* variants and corresponding G6PD activity in participants of the clinical trial and cross-sectional survey.**

| *G6PD* variant | Parasitemia | | Aparasitemia | | p-Value |
|---|---|---|---|---|---|
| | n (column percent) | Median G6PD activity in U/g Hb (IQR, minimum–maximum) | n (column percent) | Median G6PD activity in U/g Hb (IQR, minimum–maximum) | |
| **Microscopic parasitemia** | | | | | |
| Mahidol homozygous | 2 (40.0) | 2.03 (0.02–4.03, 0.02–4.03) | 56 (53.3) | 0.51 (0.16–0.78, 0.00–13.06) | 0.448 |
| Mahidol heterozygous | 1 (20.0) | 5.38 (n = 1) | 42 (40.0) | 3.72 (2.06–4.85, 0.00–14.52) | 0.348 |
| Orissa homozygous | 1 (20.0) | 0.7 (n = 1) | 6 (5.7) | 2.77 (1.21–3.17, 0.27–4.19) | 0.285 |
| Orissa heterozygous | 1 (20.0) | 6.22 (n = 1) | 0 (0.0) | n = 0 | 0.045 |
| Kalyan–Kerala homozygous | 0 (0.0) | n = 0 | 1 (1.0) | 2.07 (n = 1) | 0.955 |
| Total | 5 (100.0) | — | 105 (100.0) | — | — |
| **Submicroscopic parasitemia** | | | | | |
| Mahidol homozygous | 3 (60.0) | 0.19 (0.00–1.53, 0.00–1.53) | 53 (53.0) | 0.53 (0.18–0.77, 0.00–13.06) | 1.00 |
| Mahidol heterozygous | 2 (40.0) | 2.59 (2.06–3.11, 2.06–3.11) | 40 (40.0) | 3.81 (2.09–4.96, 0.00–15.52) | 1.00 |
| Orissa homozygous | 0 (0.0) | — | 6 (6.0) | 2.77 (1.21–3.17, 0.27–4.19) | 1.00 |
| Kalyan–Kerala homozygous | 0 (0.0) | — | 1 (1.0) | 2.07 (n = 1) | 1.00 |
| Total | 5 | — | 100 | — | — |

G6PD, glucose-6-phosphate dehydrogenase; IQR, interquartile range.

**Table 5. Significant predictors of non-normalized G6PD activity among participants with known *G6PD* variant.**

| Variable (type) | β (95% CI) | *p*-Value |
|---|---|---|
| **Contributors lowering G6PD activity** | | |
| Mahidol hemi-/homozygous (cat) | −62.6 (−71.8 to −53.5) | **<0.001** |
| Orissa hemi-/homozygous (cat) | −44.9 (−68.0 to −21.8) | **<0.001** |
| Mahidol heterozygous (cat) | −15.6 (−26.3 to −5.0) | **0.004** |
| Sex (females compared to males) (cat) | −15.5 (−23.2 to −7.8) | **<0.001** |
| Hb in g/dl (cont) | −2.0 (−4.1 to 0.0) | **0.050** |
| **Contributors increasing G6PD activity** | | |
| Microscopic parasitemia (cat) | 22.6 (9.7 to 35.4) | **0.001** |

Significant *p*-values given in bold. Model details: $F(11, 336) = 20.50$, $p < 0.001$, adjusted $R^2 = 0.3820$.

cat, categorical; cont, continuous; G6PD, glucose-6-phosphate dehydrogenase.

with G6PD activity: *G6PD* variant, female sex, and Hb concentration. Being positive for either species by microscopy, but not submicroscopic parasite carriage, was correlated positively with G6PD enzyme activity (Table 5).

## Case–control study

A total of 506 participants were enrolled in the case–control study, 253 in each arm, which were well matched (Table 6). Before enrollment, cases had been under observation for a median time of 8.7 years (IQR 8.4–8.9, range 1.3–9.5) and controls for a median time of 8.6 years (IQR 8.5–8.9, range 1.9–9.5; $p = 0.669$).

The median G6PD activity was 10.2 U/g Hb (IQR 8.7–11.8) for cases and 10.3 U/g Hb (IQR 8.8–12.2) for controls ($p = 0.323$) (Fig 4). The combined AMM for cases and controls was 10.7 U/g Hb (IQR 9.0–12.4). There was no significant difference between cases and controls in the proportion of patients categorized as G6PD normal, intermediate, or deficient ($p = 0.328$); although cases were at lower risk than controls of having a G6PD deficiency defined at the 30% threshold (OR = 0.60, 95%CI: 0.27 to 1.29) and at the 70% threshold (OR = 0.71, 95%CI: 0.41 to 1.23) this did not reach statistical significance ($p = 0.155$ and $p = 0.241$ respectively) (Fig 4; Table 7).

All but 1 of the G6PD-deficient (<30% activity) participants were genotyped ($n = 33$), and a known variant was identified in 97.0% (32/33); also, all 34 G6PD-intermediate participants were genotyped, and a variant was identified in 23.5% (8/34) of participants (S3 Table).

**Table 6. Baseline characteristics for individuals in the case–control study.**

| Variable | Cases | Controls | Difference (cases − controls) | *p*-Value |
|---|---|---|---|---|
| Females, *n* (%) | 110 (43.5%) | 110 (43.5%) | 0.0% | 1.000 |
| Mean age, years (95% CI) | 33.2 (31.1–35.4) | 33.2 (31.1–35.4) | 0.0 (−3.0 to 3.0) | 0.988 |
| Age < 5 years, *n* | 0 | 0 | 0 | 0.917 |
| Age 5–15 years, *n* | 60 | 62 | −2 | |
| Age > 15 years, *n* | 193 | 191 | 2 | |
| Mean body weight, kg (95% CI) | 45.4 (43.9–47.0) | 45.6 (44.0–47.3) | 0.2 (−2.5 to 2.0) | 0.850 |
| Mean body height, cm (95% CI) | 152.1 (150.5–153.7) | 151.7 (150.0–153.4) | 0.4 (−2.0 to 2.7) | 0.738 |
| Mean body temperature, ˚C (95% CI) | 36.0 (35.9–36.0) | 36.0 (35.9–36.0) | 0.0 (0.0 to 0.1) | 0.688 |
| Mean Hb concentration, g/dl (95% CI) | 14.3 (14.0–14.6) | 14.4 (14.2–14.7) | −0.1 (−0.5 to 0.3) | 0.622 |
| Severe anemia (Hb < 7 g/dl), *n* (%) | 3 (2.7%) | 0 | 3 | 0.249 |
| Median person-years of observation (IQR, range) | 8.67 (8.44–8.88, 1.33–9.50) | 8.62 (8.45–8.88, 1.91–9.50) | −0.03 (−0.22 to 0.17) | 0.669 |

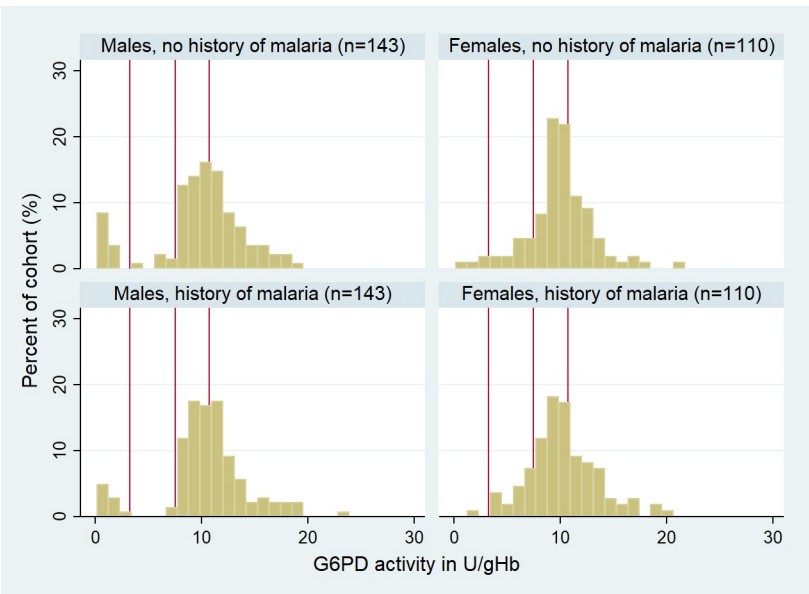

**Fig 4. Distribution of G6PD activity stratified by sex in cases and controls.** Red vertical lines indicate 30%, 70%, and 100% G6PD activity of the adjusted male median. G6PD, glucose-6-phosphate dehydrogenase.

**Table 7. Distribution of G6PD-deficient, -intermediate, and -normal participants among cases and controls.**

| G6PD category | Cases, n (%) | Controls, n (%) | Total, n (%) |
|---|---|---|---|
| Deficient | 13 (5.1) | 21 (8.3) | 34 (6.7) |
| Intermediate | 16 (6.3) | 18 (7.1) | 34 (6.7) |
| Normal | 224 (88.5) | 214 (84.6) | 438 (86.6) |
| Total | 253 | 253 | 506 |

G6PD, glucose-6-phosphate dehydrogenase.

**Table 8. Significant predictors of non-normalized G6PD activity among cases and controls.**

| Variable (type) | β (95% CI) | p-Value |
|---|---|---|
| Mahidol hemi-/homozygous (cat) | −76.2 (−91.5 to −60.9) | <**0.001** |
| Mediterranean heterozygous (cat) | •−52.8 (−95.2 to −10.3) | **0.016** |
| Orissa hemi-/homozygous (cat) | −39.7 (−65.5 to −13.9) | **0.011** |

Significant p-values given in bold. Model details: $F(7, 60) = 27.91$, $p < 0.001$, adjusted $R^2 = 0.7377$, $n = 68$.
cat, categorical variable; G6PD, glucose-6-phosphate dehydrogenase.

The median non-normalized G6PD activity was 147.4 U/dl (IQR 123.0–171.5). In a multi-variable regression model ($F[7, 60] = 27.91$, $p < 0.001$, adjusted $R^2 = 0.7377$, $n = 68$), being hemi- or homozygous for the Mahidol or Orissa variant and being heterozygous for the Mediterranean variant were found to be significant predictors of non-normalized G6PD activity, but Hb ($p = 0.169$), sex ($p = 0.693$), and a history of malaria ($p = 0.725$) were not significant predictors (Table 8).

## Discussion

Our study highlights a significant difference in G6PD activity between patients with clinical malaria and asymptomatic individuals who were either aparasitemic or had very low-level parasitemia that was only detected by PCR. This difference could not be accounted for by a protective effect against malaria from low G6PD activity [8]. Although individuals with a history of malaria were less likely to be G6PD deficient or G6PD intermediate, the ORs were only 0.6 and 0.7, respectively, and did not reach statistical significance.

Within an individual, G6PD activity varies with the age of the RBC population, with younger RBCs, reticulocytes, and nucleated normoblasts having higher G6PD activity [13,34–36]. Malaria induces hemolysis, to which the host responds through reactive erythropoiesis, increasing the number of reticulocytes and, in severe cases, normoblasts within the bloodstream [36–39]. As a consequence, the mean age of the host RBC population decreases, while the mean G6PD activity increases. It is therefore plausible that G6PD activity rises during or shortly after an episode of acute malaria [40].

The WHO malaria treatment guidelines recommend that patients with malaria who have G6PD enzyme activity less than 30% should not be treated with a standard 14-day regimen of PQ, but rather PQ should be administered weekly for 8 weeks [33]. These guidelines were formulated following studies suggesting that, except for the Mediterranean variant, PQ-induced hemolysis is self-limiting and that an initial dose of PQ would lead to a replacement of older RBCs with younger cells, conferring transitional low-grade resistance against PQ-induced hemolysis [32,33,40–43]. Acute malaria may have a similar effect.

The hemolytic risk associated with PQ rises as G6PD activity falls [43]. G6PD-deficient individuals with activity below 30% can be identified with qualitative diagnostics that can be used in remote settings [44]. Heterozygous females with intermediate G6PD activity (30% to <70%) are also at a significant risk of PQ-induced hemolysis, but their diagnosis requires a quantitative enzyme assay [45]. Our analysis highlights that, compared to females with subpatent parasitemia or aparasitemia, females with microscopic parasitemia were 14 times less likely to be G6PD intermediate or deficient; the corresponding OR for males was 3.6. We hypothesize that this is due to preferential parasite-induced hemolysis of G6PD-deficient RBCs in mosaic heterozygous females. If confirmed, our findings suggest that the risk of PQ-induced hemolysis in heterozygous females with malaria would be considerably less than that predicted by cross-sectional surveys of healthy populations.

Our study has some important limitations. We relied on non-contemporaneous collection of specimens from a prospective, single-arm clinical trial and cross-sectional survey, rather than clinical observation of the same patients over multiple time points with and without malaria. However, when comparing results from laboratory quality control testing, results did not differ significantly, suggesting that the results of the studies are comparable.

In the case–control study, participants were tested using a combination RDT for both *P. falciparum* and *P. vivax*, and this is known to have lower sensitivity for identifying patients with *P. vivax* malaria [46]. It is also possible that some participants had malaria that was not detected due to treatment-seeking outside of the surveillance system. However, patients must pay for services and drugs provided at private clinics, pharmacies, and, to some extent, public hospitals; it is unlikely that many participants would prefer this option over free treatment delivered at home as part of the DSS. A subset of 100 participants from each case and control arm were included in a separate study of the perceptions of malaria [47]. A total of 6 participants from the control arm recalled having had malaria previously, whereas only 1 case could not recall any previous episodes of malaria. While these findings are vulnerable to recall and response bias, they suggest that the majority of participants were categorized correctly.

There was a tendency for individuals without a history of malaria to be G6PD intermediate or deficient, as has been observed by others [8,48–50]. A recent study among more than 17,000 infants and neonates in Kenya found the odds in heterozygous females of being admitted to hospital with a malaria-positive microscopy slide to be lower (OR 0.79, 95% CI 0.68–0.93, $p$ = 0.004) compared to G6PD-normal individuals; a recent article from Afghanistan reported an even greater protective effect against *P. vivax* malaria for the Mediterranean variant (76% in hemizygous males and 55% in females) [51,52]. In our analysis the protective effect was modest and did not reach statistical significance. If a protective effect of G6PDd against malaria in Bangladesh was present, it is likely to be a minor contributor to G6PD status and would not have accounted for asymptomatic and parasite-free individuals being 4- to 9-fold more likely to be diagnosed as G6PD deficient in the clinical trial and cross-sectional survey.

More than 30% of participants were genotyped against a panel of *G6PD* variants known to be prevalent in the study area [4,53,54]. However, a genetic variant could not be identified in a large proportion of phenotypically G6PD-deficient individuals, perhaps indicative of the genetic diversity in the area [20]. Only patients with known variants were included in the regression models, and, accordingly, these were restricted to 25% of all study participants, which could have affected their validity. Reassuringly, in the multivariable models, the phenotypic observations highlighted that malaria was a significant predictor of phenotypic G6PD activity, whereas previous history of malaria was not.

In conclusion, our study highlights the potential impact of acute malaria on G6PD status, which may influence treatment allocation of 8-aminoquinoline drugs. If our findings are confirmed prospectively, 8-aminoquinoline-based radical cure may be safer than assumed in patients with clinical malaria, compared to individuals who have submicroscopic parasitemia or who are aparasitemic.

## Supporting information

**S1 Fig. Distribution of G6PD activity (in U/g Hb) in patients with and without peripheral parasitemia.** Red vertical lines indicate 30%, 70%, and 100% G6PD activity of the adjusted male median. 100% G6PD activity = 7.4 U/g Hb.
(TIF)

**S1 Table. Protocol names and ethics identifiers.**
(DOCX)

**S2 Table. Distribution of G6PD activity among genotyped participants.**
(DOCX)

**S3 Table. Distribution of G6PD variants among cases and controls and corresponding G6PD activity.**
(DOCX)

**S1 File. STROBE table.**
(DOCX)

**S2 File Database. Control results of clinical trial and cross-sectional survey.**
(XLS)

**S3 File Database. Clinical trial and cross-sectional survey.**
(XLS)

**S4 File Database. Case–control study.**
(XLS)

## Acknowledgments

We would like to thank all participants who agreed to participate, and all staff involved in making this work happen. icddr,b is grateful to the governments of Bangladesh, Canada, Sweden, and the UK for providing core/unrestricted support.

## Author Contributions

**Conceptualization:** Benedikt Ley.

**Data curation:** Mohammad Sharif Hossain.

**Formal analysis:** Benedikt Ley.

**Investigation:** Nusrat Jahan, Fatema Tuj Johora, Cristian Koepfli.

**Methodology:** Benedikt Ley.

**Project administration:** Jutta Marfurt, Ching Swe Phru.

**Supervision:** Mohammad Shafiul Alam, Wasif Ali Khan.

**Visualization:** Benedikt Ley.

**Writing – original draft:** Benedikt Ley.

**Writing – review & editing:** Benedikt Ley, Mohammad Shafiul Alam, Mohammad Golam Kibria, Jutta Marfurt, Ching Swe Phru, Jenifar Quaiyum Ami, Kamala Thriemer, Sarah Auburn, Cristian Koepfli, Wasif Ali Khan, Ric N. Price.

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
