## [Editor Report · Decision Letter 0]

18 Sep 2020

Dear Dr Ley, 

Thank you for submitting your manuscript entitled "Variability of Glucose-6-Phosphate Dehydrogenase activity in individuals with and without malaria in the Chittagong Hill Tracts of Bangladesh" for consideration by PLOS Medicine.

Your manuscript has now been evaluated by the PLOS Medicine editorial staff and I am writing to let you know that we would like to send your submission out for external peer review.

Kind regards,

Artur Arikainen,

Associate Editor

PLOS Medicine

---

## [Decision Letter · Decision Letter 1]

11 Dec 2020

Dear Dr. Ley,

Thank you very much for submitting your manuscript "Variability of Glucose-6-Phosphate Dehydrogenase activity in individuals with and without malaria in the Chittagong Hill Tracts of Bangladesh" (PMEDICINE-D-20-04491R1) for consideration at PLOS Medicine. 

Your paper was evaluated by a senior editor and discussed among all the editors here. It was also discussed with an academic editor with relevant expertise, and sent to three independent reviewers, including a statistical reviewer (#r1). The reviews are appended at the bottom of this email and any accompanying reviewer attachments can be seen via the link below:

[LINK]

In light of these reviews, I am afraid that we will not be able to accept the manuscript for publication in the journal in its current form, but we would like to consider a revised version that addresses the reviewers' and editors' comments. Obviously we cannot make any decision about publication until we have seen the revised manuscript and your response, and we plan to seek re-review by one or more of the reviewers. 

We expect to receive your revised manuscript by Jan 01 2021 11:59PM. Please email us (plosmedicine@plos.org) if you have any questions or concerns.

We look forward to receiving your revised manuscript. 

Sincerely,

Emma Veitch, PhD

PLOS Medicine

On behalf of Richard Turner PhD, Senior Editor, 

PLOS Medicine

plosmedicine.org

*PLOS Medicine style for the title is normally to include some designation of the study design for the methodological approach (usually after a colon following the study question/objective) - here I appreciate that is tricky because three groups are involved (trial, x-sectional and case-control). If the authors are amenable however we'd suggest stating that in the title subtitle (eg, "Variability of Glucose-6-Phosphate Dehydrogenase activity in individuals with and without malaria: analysis of trial, cross-sectional and case-control data" for example.

*There is some confusion in the abstract where initially the authors state they have analysed patient groups per the above frameworks (trial, cross-sectional and case-control data) but later on the authors suggest they are analysing a prospective cohort. For consistency the authors need to be clear, and also to help the readers follow which of the groups they are referring to. 

*The Author Summary section would need some stylistic changes to fit our usual format, please check the guidelines here: https://journals.plos.org/plosmedicine/s/revising-your-manuscript#loc-author-summary and note this should be 2-3 single sentence bullet points for each of the sections; see also an example at: https://journals.plos.org/plosmedicine/article?id=10.1371/journal.pmed.1002416

*Where the authors mention the Ley et al accepted paper in PLOS NTD's, we'd suggest instead noting the accepted paper in the reference list as for the other papers - give the author list, paper title, and then PLOS NTD's "accepted for publication". PLOS Med reference policy allows for inclusion in the reference list of accepted publications (https://journals.plos.org/plosmedicine/s/submission-guidelines#loc-references) and it will be more straight forward to simply call out the paper with a citation number and then point the readers to the reference in the reflist. If a DOI is available from NTDs please include that, and if the NTD paper gets a formal citation before your PLOS Medicine paper gets a final decision, you can update the citation in the reflist.

*For the analysis of trial data that is from a PLOS ONE published trial, we'd also suggest including the clinicaltrials.gov ID number directly in the manuscript text, as well as the reference citation - https://clinicaltrials.gov/ct2/show/NCT02389374

*Please clarify whether the analytical approach followed here corresponds to one laid out in a prospective protocol or analysis plan? Please state this (either way) early in the Methods section.

*Although this is complex because the paper uses data from three separate participant groups, we'd suggest aiming to ensure the study is reported according to the STROBE guideline, and include the completed STROBE checklist as Supporting Information. The important thing here would be to ensure that the elements of the guideline are covered in the reporting in the paper, rather than to try to fill out the checklist in a way that reflects all three study populations. We'd also suggest including this statement (or something along these lines) in the methods section of the paper - "This study is reported as per the Strengthening the Reporting of Observational Studies in Epidemiology (STROBE) guideline (S1 Checklist)." The STROBE guideline can be found here: http://www.equator-network.org/reporting-guidelines/strobe/

Comments from the reviewers:

Reviewer #1: See attachment

Michael Dewey

Reviewer #2: 

G6PD activity is dependent on genotype and RBC's age. The authors tried to clarify the real time enzymatic activity in the individuals who are exposed to acute malaria using a DSS cohort and a clinical efficacy trial in the endemic area of Bangladesh. The authors found the G6PD activity showed a significant increase in malaria positive patients compared with malaria negative or asymptomatic PCR positives. 

Combined with a case control study of past malaria history positives, the authors concluded that the G6PD activity increased after the hemolysis of parasitized old RBCs which is independent from genotype origin. Field study and related laboratory protocol including enzymatic estimation and genotyping are reasonable. However, the conclusion could not be completely supported by their findings. To make more convincing discussion and recommendation, several major points should be clarified.

Major comment

1. Case-control study needs more details because this is essential to lead to their conclusion that malaria infection increased the enzyme activity, For the reviewer, there are a lot of questions about the criteria of malaria history positive or negative. 

Even the protective effect of G6PDd is controversial in Asia and Africa especially against different species such as P. falciparum and P. vivax. Please describe how many Pf and Pv.

The endemicity of malaria is not clear there. Please describe how many positive cases observed in the cohort population. Also the authors should explain the seasonality and annual changes of the endemicity.

2. The comparison between G6PD low, intermediate and normal groups for their malaria infection strongly suggested a protective effect of low enzymatic activity which is opposite to the description at page 21, line 356. For the reviewer, the protective effect looks very clear. However, the authors denied this possibility. Please explain the reason why without mentioning the case control study.

3. Regarding genetic polymorphism, the Mahidol variant looked a major one. Please describe more about the genotype information about female population in the Figure 3 separating by 30,70,100% activity.

Minor comment

1. In the title, is it necessary to use Variability?

2. Please explain the relationship between acute malaria infection time course and PQ treatment.

3. Please cite any reference to give an evidence of increased G6PD activity in experimental malaria model or in human study. Or is it possible to show any information about the enzymatic activity of the same individual when infected or not?

4. Page 21, line 378. I need a detailed description about this odds. It is very difficult for me to understand.

Reviewer #3: 

Accept with suggestions for a few minor revisions

The paper by Benedikt Ley et al. of the Variability of Glucose-6-Phosphate Dehydrogenase activity in individuals with and without malaria in the Chittagong Hill Tracts of Bangladesh:

The authors measured G6PD activities among 

- individuals with and without malaria and found G6PD activities to be significantly higher among malaria patients. 

- individuals enrolled healthy with and without a history of malaria to assess whether people with low G6PD activities were less likely to contract malaria. They found thatG6PD activities did not differ significantly.

They conclude that G6PD activity increases during a malaria infection and the observed clinically relevant difference is not due to people with lower G6PD activities being less likely to contract malaria. 

 The title and abstract are appropriate for the content of the text. Furthermore, the article is well constructed and analysis was well performed.

The authors demonstrated that their study highlights the potential impact of acute malaria on G6PD status, which may influence treatment allocation of an 8-aminoquinoline drug. this a good alternative because the cure may be safer than assumed in patients with clinical malaria, compared 

The article highlights important data to know, however, we have some suggestions.

1. The document: need a few minor amendment (formatting, Typos, spelling, grammar, and phrasing issues). I have propose some suggestions. 

2. The authors should clarify and give more explanation in the discussion sections: paragraph bellow:

"During laboratory quality control testing, there was a significant difference in the enzyme activity of controls between the efficacy study and the cross-sectional survey that may have influenced the direct comparison of results

3. The authors should confirm if their findings indicate a substantial variation in G6PD measurements between the different methods used and different studies. We will suggest that the authors give more explication because caution is advised in comparing findings based on absolute G6PD activity measurements across studies.

4. The authors should give if possible the mean temperature and mean precipitation during the different periods of three studies (The main climatic factors that influence the transmission of malaria are temperatures and precipitation)

5. The authors have identified much literature to relates to this research. However, we suggest that the authors using also the below some publications to explain the introduction

1) Pfeffer DA, Ley B, Howes RE, Adu P, Alam MS, Bansil P, et al. (2020) Quantification of glucose-6-phosphate dehydrogenase activity by spectrophotometry: A systematic review and metaanalysis. PLoS Med 17(5): e1003084. https://doi. org/10.1371/journal.pmed.1003084

2) Kießling et al. Malar J (2018) 17:358 https://doi.org/10.1186/s12936-018-2510-3

3) Emilie S. Badoum, Samuel S. Sermé, Jean B. Yaro, Sam. A Coulibaly, Désiré Kargougou, Amidou Diarra, Amidou Z Ouédraogo, Lankoande Malik, Issa Nébié, Issiaka Soulama, Alphonse Ouédraogo, Alfred B. Tiono, Yves Traoré, Sodiomon B. Sirima, Edith C. Bougouma* Abnormalities of hemoglobin and Glucose-6-Phosphate-Dehydrogenase deficiency in children with uncomplicated malaria and living in Banfora and Saponé, two different malaria setting of Burkina Faso International Journal of TROPICAL DISEASE & Health 37(3): 1-10, 2019; Article no.IJTDH.50379 ISSN: 2278-1005, NLM ID: 101632866

[LINK]

---

## [Decision Letter · Decision Letter 2]

15 Feb 2021

Dear Dr. Ley,

Thank you very much for re-submitting your manuscript "Glucose-6-Phosphate Dehydrogenase activity in individuals with and without malaria:  analysis of clinical trial, cross-sectional and case-control data from Bangladesh" (PMEDICINE-D-20-04491R2) for review by PLOS Medicine.

I have discussed the paper with my colleagues and the academic editor and it was also seen again by two reviewers. I am pleased to say that provided the remaining editorial and production issues are dealt with we are planning to accept the paper for publication in the journal.

[LINK]

We look forward to receiving the revised manuscript by Feb 22 2021 11:59PM.   

Sincerely,

Beryne Odeny, 

Associate Editor 

PLOS Medicine

plosmedicine.org

Requests from Editors:

Thank you for addressing the previously stated issues. Before we proceed, please address the following issues:

• Was the clinical trial registered in a recognized database?

• At key points in the paper, i.e., in the abstract, methods section and discussion, please adapt the description "clinical trial" to provide slightly more information on the study design, e.g., "prospective, single-arm clinical trial"

• Please reduce the length of your abstract, if possible.

• Please quote study dates in your abstract; along with number of participants in the individual studies and aggregate demographic details

• At line 98, it may be that "lyonization" should take an initial capital, or the alternative term used.

• Please remove trademarks from your manuscript, e.g. at line 159

• Please remove information on funding from the title page and end of the main text. This information will appear in the article metadata upon publication, via information in the submission form

• Please hyphenate "case-control" consistently.

• Please add the following statement, or similar, to the Methods: "This study is reported as per the Strengthening the Reporting of Observational Studies in Epidemiology (STROBE) guideline (S1 Checklist)."

• Please indicate in the figure caption the meaning of the red vertical lines in Fig. 3 and Fig. 4

• Please define the abbreviations in all tables and figures, including: G6PD, RDT, PCR, IQR

• Please provide titles and legends for all figures (including those in Supporting Information files).

• Please use the "Vancouver" style for reference formatting and see our website for other reference guidelines https://journals.plos.org/plosmedicine/s/submission-guidelines#loc-references . Please note that citations should be in square brackets.

• Thank you for providing your [CONSORT/PRISMA/STROBE] checklist. Please replace the page numbers with paragraph numbers per section (e.g. "Methods, paragraph 1"), since the page numbers of the final published paper may be different from the page numbers in the current manuscript.

• Line 409 -410, the term "trend" is used to refer to a nonsignificant P value. The term trend should be used only when the test for trend has been conducted. Please revise accordingly.

• Please style reference call outs as follows throughout the text, noting the absence of spaces within the square brackets: "... countries [1,2]."

• In the reference list, please ensure that all citations have full access details, e.g., reference 1.

• Please abbreviate journal names consistently, e.g., "PLoS ONE"; "PLoS Med.".

Comments from Reviewers:

Reviewer #1: The authors have addressed my points.

Michael Dewey

Reviewer #2: The authors reacted to reviewers' comments properly and the revised paper is acceptable for publication.

[LINK]

---

## [Editor Report · Decision Letter 3]

1 Mar 2021

Dear Dr Ley, 

On behalf of my colleagues and the Academic Editor, Kenji Hirayama, I am pleased to inform you that we have agreed to publish your manuscript "Glucose-6-Phosphate Dehydrogenase activity in individuals with and without malaria:  analysis of clinical trial, cross-sectional and case-control data from Bangladesh" (PMEDICINE-D-20-04491R3) in PLOS Medicine.

PRESS

Sincerely, 

Beryne Odeny 

Associate Editor 

PLOS Medicine